# DTVLT: A Multi-modal Diverse Text Benchmark for Visual Language Tracking Based on LLM

## Abstract

Visual language tracking (VLT) has emerged as a cutting-edge research area, harnessing linguistic data to enhance algorithms with multi-modal inputs and broadening the scope of traditional single object tracking (SOT) to encompass video understanding applications. Despite this, most VLT benchmarks still depend on succinct, human-annotated text descriptions for each video. These descriptions often fall short in capturing the nuances of video content dynamics and lack stylistic variety in language, constrained by their uniform level of detail and a fixed annotation frequency. As a result, algorithms tend to default to a "memorize the answer" strategy, diverging from the core objective of achieving a deeper understanding of video content. Fortunately, the emergence of large language models (LLMs) has enabled the generation of diverse text. This work utilizes LLMs to generate varied semantic annotations (in terms of text lengths and granularities) for representative SOT benchmarks, thereby establishing a novel multi-modal benchmark. Specifically, we (1) propose a new **v**isual **l**anguage **t**racking benchmark with **d**iverse **t**exts, named **DTVLT**, based on five prominent VLT and SOT benchmarks, including three sub-tasks: short-term tracking, long-term tracking, and global instance tracking. (2) We offer four granularity texts in our benchmark, considering the extent and density of semantic information. This is achieved through DTLLM-VLT, a method for generating high-quality, diverse text by leveraging the extensive knowledge base of LLMs to produce descriptions rich in world knowledge. We expect this multi-granular generation strategy to foster a favorable environment for VLT and video understanding research. (3) We conduct comprehensive experimental analyses on DTVLT, evaluating the impact of diverse text on tracking performance and hope the identified performance bottlenecks of existing algorithms can support further research in VLT and video understanding.

## 1 Introduction

Single object tracking (SOT) is a pivotal task in computer vision, designed to follow a single moving object throughout a video sequence. Researchers have observed that the effectiveness of most trackers often diminishes when tracking objects in lengthy videos with intricate content. Moreover, relying solely on visual cues significantly hinders the flexibility of these trackers.

Consequently, there has been a pronounced trend in recent studies to integrate semantic annotations into SOT, leading to the development of the visual language tracking (VLT) task. This new task is advantageous as it extends the potential applications of SOT, including advancements in video understanding. Utilizing natural language in place of bounding boxes (BBox) provides a more user-friendly and intuitive alternative. This method facilitates more precise representations of objects, encompassing both their spatial positioning and intricate semantic attributes, thereby enhancing the efficacy of the tracking process.

When defining the VLT task, researchers integrate text annotations from two primary perspectives: (1) **Short text annotations.** Representative VLT benchmarks such as OTB99_Lang (Li et al. (2017)), TNL2K (Wang et al. (2021)), and LaSOT (Fan et al. (2019; 2021)) primarily employ short text. This straightforward approach is clear and easy to understand, aiding in the learning and comprehension by

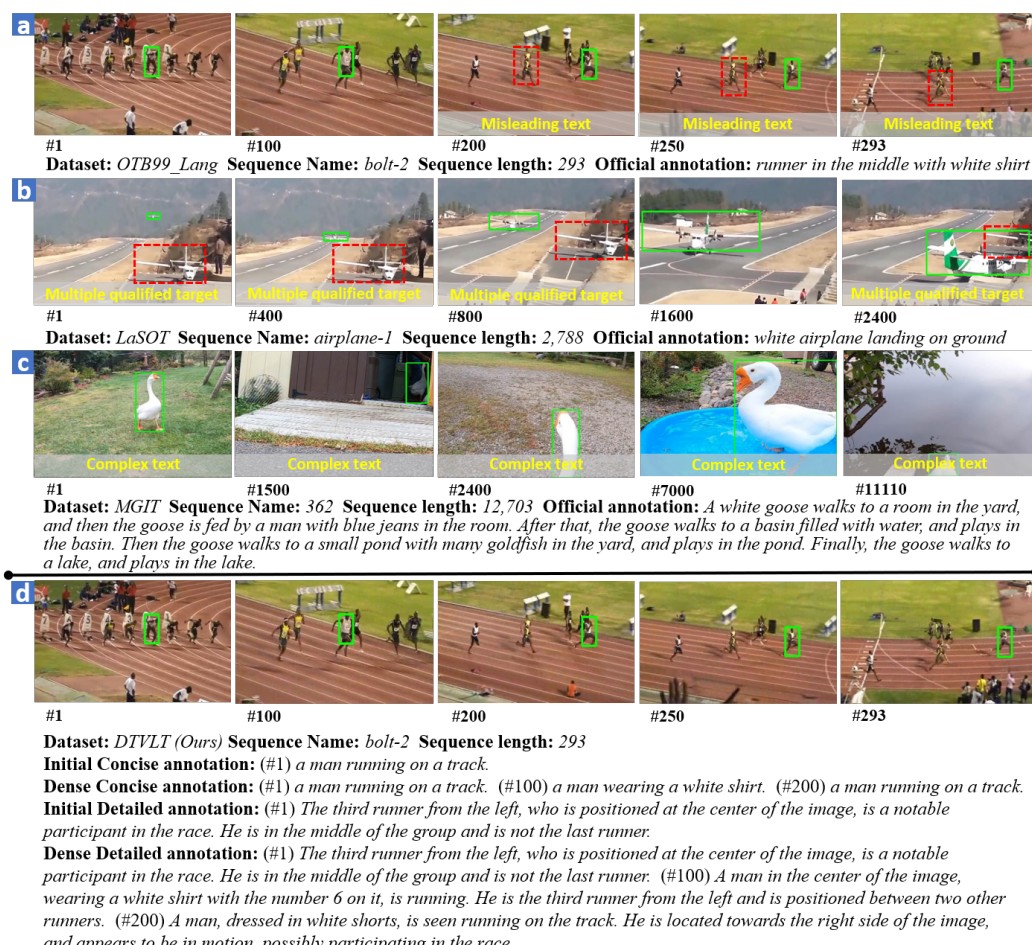

Figure 1: Comparison of DTVLT and other VLT benchmarks. (a-c) Examples of video content and semantic descriptions on OTB99_Lang (Li et al. (2017)), LaSOT (Fan et al. (2019)), and MGIT (Hu et al. (2023a)). The green bounding box (BBox) indicates ground truth, while the red dashed BBox indicates other objects that satisfy the semantic description. (a) and (b) are sequences with simple narrative content. And their semantic annotations mainly describe the first frame, which may misguide the algorithm bacause of misleading text and multiple qualified target. In the VLT task, if the error caused by incorrect text accumulates for the tracker, it will have an irreversible impact on the tracking results. (c) in MGIT has such complex text that they are not conducive to algorithmic learning. (d) An example of the multi-granular generation strategy used by DTVLT. We provide more diverse concise and detailed descriptions for each hundred frames of the object to be tracked, covering five representative datasets across three mainstream tracking tasks. The term "# xx" represents the frame ID. Compared to existing benchmarks, the generated text provides more prosperous and flexible information to portray long videos.

VLT trackers. However, these methods are susceptible to vague semantic descriptions and potential ambiguities. For instance, as depicted in Fig. 1 (a) and (b), the description captures only the object's initial state. As the object moves, the positional constraint in the semantic information can become misleading, making the semantic descriptions restrictive over time. (2) **Long text annotation.** MGIT (Hu et al. (2023a)) adopts a multi-granular semantic annotation strategy aimed at providing more precise semantic descriptions. This method stands out from other benchmarks with two key features: extended text lengths and periodic updates, transitioning from simple to dense and detailed descriptions. Nonetheless, this approach encounters challenges such as the time-consuming nature of text annotation and the necessity for algorithms capable of robust text processing and multi-modal alignment. As shown in Fig. 1 (c), the text in MGIT can be excessively lengthy and complex.

Clearly, while the intent of these studies is to extend the SOT task into a multi-modal one to improve tracking performance, the singular granularity used in most research not only impedes algorithms from

achieving the desired results but also complicates VLT research. Therefore, a superior approach to constructing a VLT benchmark would be to move beyond offering a mere natural language description for short videos. Instead, it would involve devising a systematic method to supply multi-granular texts that support trackers in understanding various video contents.

Offering a variety of environmental texts—including short, long, sparse, and dense formats—and evaluating algorithm performance across these descriptions allows us to effectively identify the strengths and weaknesses of methods under various semantic granularities. This insight can guide the improvement of multi-modal algorithms. What excites us is the potential of large language models (LLMs) to aid in reaching this objective. By integrating the LLM seamlessly into the process of text generation, we can create a diverse multi-modal environment that is favorable for VLT research.

Our work is motivated by the aforementioned considerations and aims to construct a new VLT benchmark named DTVLT. This benchmark leverages the DTLLM-VLT (Li et al. (2024a)) method, which employs LLM to generate a wide variety of texts for tracking datasets. Specifically, we integrate text length and generation density to create four distinct levels of granularity. With this framework, we have selected a range of VLT trackers for experimental analysis to assess how diverse texts affect algorithmic performance. The experimental results not only illustrate that this diversified setting can support detailed evaluation and analysis of algorithmic capabilities but also indicate the potential for future improvements in the multi-modal learning capabilities of trackers by using generated data.

**Contributions.** (1) We propose a new VLT benchmark named DTVLT based on five prominent VLT and SOT benchmarks including three tracking tasks: short-term tracking, long-term tracking, and global instance tracking. (2) We offer four granularity combinations for our benchmark, considering the extent and density of semantic information using DTLLM-VLT, which leverages LLM to generate diverse high-quality language descriptions. We expect this multi-granular generation strategy can provide a favorable environment for VLT and video understanding research. (3) We conduct comprehensive experimental analyses, evaluating the impact of diverse text on tracking performance and hope the explored performance bottlenecks of existing algorithms can support further VLT research.

## 2 RELATED WORK

**Single Object Tracking Benchmark.** The SOT task involves initializing and tracking a specific object within a video sequence. It starts by identifying the object through its bounding box (BBox) in the first frame and then continues to track and locate the object in subsequent frames. Since 2013, several benchmarks, such as OTB (Wu et al. (2013; 2015)) and VOT (Kristan et al. (2016); Bibliographie Goecke et al. (2013); Kristan et al. (2015; 2018; 2019)), have been developed to provide standardized datasets and scientific evaluation mechanisms for SOT research. However, with the progress in deep learning techniques, these short-term and small-scale benchmarks have faced difficulties in sufficiently supporting data-driven trackers. This has led researchers to create larger-scale datasets like GOT-10k (Huang et al. (2021)) and TrackingNet (Muller et al. (2018)). Some work has also focused on SOT in drone scenarios, such as BioDrone (Zhao et al. (2023b)), a vision benchmark for SOT based on bionic drones and WebUAV-3M (Zhang et al. (2022)). More recently, researchers introduced the global instance tracking task along with a new benchmark called VideoCube (Hu et al. (2023b)), allowing the tracking of arbitrary moving objects in various types of videos. To scientifically assess tracker performance under different challenging conditions, researchers have also introduced SOTVerse (Hu et al. (2024)), a user-defined space for the SOT task.

**Visual Language Tracking Benchmark.** Over the past few decades, visual benchmarks have seen considerable development, yet benchmarks that incorporate semantic information, known as VLT benchmarks, have only recently become prominent. OTB99_Lang (Li et al. (2017)) is notable for being the first VLT benchmark, augmenting the sequences from the OTB100 (Wu et al. (2015)) benchmark with natural language descriptions. However, the limited scale of the dataset has hindered the broader acceptance of the VLT task. Following this, the introduction of LaSOT (Fan et al. (2019; 2021)), a multi-modal benchmark for long-term tracking, represented a major advancement. In the same year, researchers launched the TNL2K (Wang et al. (2021)) benchmark, which aimed to improve the flexibility and precision of object tracking through text descriptions. Recently, researchers have proposed a novel multi-modal benchmark called MGIT (Hu et al. (2023a)), which introduces a

Table 1: Comparison of current datasets for object tracking. DTVLT is the first comprehensive VLT benchmark using LLM to provide multi-granularity diverse semantic information, covering five mainstream tracking datasets across three tracking tasks. "STT", "LTT" and "GIT" refer to Short-term Tracking, Long-term Tracking and Global Instance Tracking.

| Dataset | Video number | Min frame | Mean frame | Max frame | Total frames | Tracking task | Text Annotation | | | |
|---------|--------|-----------|------------|-----------|--------------|---------------|-------------|-----------------|-------------|------|
| | | | | | | | Granularity | Sentence Number | Word Number | Tool |
| **OTB99_Lang** | 99 | 71 | 590 | 3,872 | 59K | STT | 1 | 99 | 358 | Human |
| **GOT-10k** | 10,000 | 29 | 149 | 1,418 | 1.5M | STT | 0 | 0 | 0 | - |
| **LaSOT** | 1,400 | 1,000 | 2,506 | 11,397 | 3.52M | LTT | 1 | 1,400 | 9,842 | Human |
| **TNL2K** | 2,000 | 21 | 622 | 18,488 | 1.24M | STT | 1 | 2,000 | 10,098 | Human |
| **MGIT** | 150 | 4,008 | 14,920 | 29,834 | 2.03M | GIT | 3 | 1,753 | 77,652 | Human |
| **DTVLT (Ours)** | 13,134 | 21 | 611 | 29,834 | 8.17M | STT & LTT & GIT | 4 | 240.8K | 5.2M | LLM |

[1]"K" stands for "thousand" and "M" stands for "million".

multi-granular annotation approach (Li et al. (2024b)). VastTrack (Peng et al. (2024)) facilitates the development of more general visual tracking via encompassing abundant classes and videos.

**Algorithms for Visual Language Tracking.** VLT is a burgeoning multi-modal task that seeks to enhance tracking by utilizing both linguistic descriptions and initial template. Most current VLT trackers (Guo et al. (2022); Wang et al. (2023); Zhao et al. (2023a); Li et al. (2022); Feng et al. (2019); Wang et al. (2018); Feng et al. (2020); Zhang et al. (2024; 2023)) operate on the principle of similarity-matching, using language descriptions and template patches to pinpoint the most analogous object within the search frame. Among these, SNLT (Feng et al. (2021)) stands out with its adaptable language-based region proposal network, which boosts tracking precision by using a dynamic aggregation mechanism. On the other hand, MMTrack (Zheng et al. (2023)) offers a streamlined and potent approach to tracking, viewing the VLT task as a series of token generation. Certain VLT trackers have started to incorporate temporal data to build a more dynamic reference. For example, GTI (Yang et al. (2021)) and AdaSwitcher (Wang et al. (2021)) recognize objects by combining tracking and localization results at each time step. JointNLT (Zhou et al. (2023)) also moves in this direction by incorporating temporal information as queries during the prediction phase. UVLTrack (Ma et al. (2024)) design a modality unified feature extractor and propose a multi-modal contrastive loss. QueryNLT (Shao et al. (2024)) ensures spatio-temporal consistency by leveraging historical visual information to improve tracking performance.

Most benchmarks for VLT offer a single natural language description per video, with text annotations that are either overly simplistic or excessively complex. These inconsistencies impede the evaluation of algorithms and the understanding of video content for VLT trackers. Additionally, all these studies provide semantic information through manually annotated data, which is a lengthy and resource-intensive process. Fig. 1 suggests that a more scientific approach is also necessary for delivering high-quality semantic information. These limitations have led us to propose DTVLT, the first comprehensive VLT benchmark using LLM to provide multi-granularity diverse semantic information, with the aim of creating a more flexible and comprehensive environment for VLT and video understanding research.

## 3 CONSTRUCTION OF DTVLT

### 3.1 DATA COLLECTION

We have chosen five notable datasets—OTB99_Lang (Li et al. (2017)), GOT-10k (Huang et al. (2021)), LaSOT (Fan et al. (2019)), TNL2k (Wang et al. (2021)), and MGIT (Hu et al. (2023a))—to build DTVLT. (For further details, please refer Appendix A.2.) GOT-10k stands as a traditional SOT benchmark. OTB99_Lang and LaSOT are enhancements of traditional SOT benchmarks, incorporating additional language annotations. TNL2k is a benchmark created specifically for the VLT task. It is worth noting that OTB99_Lang, GOT-10k, and TNL2k are considered representative datasets for short-term tracking, primarily offering a text for the first frame of each sequence. LaSOT, on the other hand, represents long-term tracking. Its textual annotations focus solely on describing the appearance of the target object, without including information about relative positions. MGIT

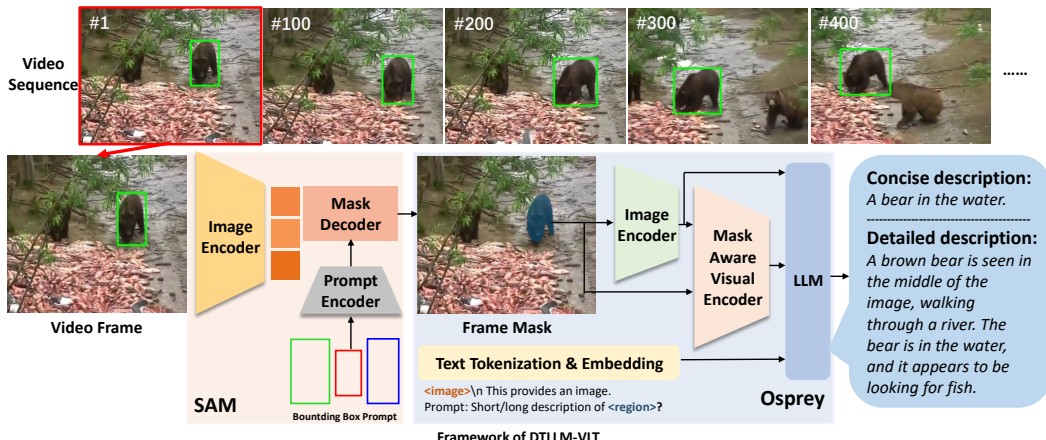

Figure 2: The pipeline of text generation for DTVLT based on DTLLM-VLT, which can provide dense concise/detailed text generation based on given video frames and BBox of object.

introduces a new, large-scale benchmark for global instance tracking. The text annotations for each sequence employ a multi-granular annotation strategy.

## 3.2 GENERATION TOOL

Traditional datasets for VLT are dependent on manual annotations. This process is costly, operates at a single annotation granularity, and is not suitable for annotating large volumes of data. To overcome these challenges, we have developed DTLLM-VLT (Li et al. (2024a)), a method capable of producing extensive and diverse text generation based on LLM. The pipeline of text generation for DTVLT is illustrated in Fig. 2. By taking video frames and the BBox of objects as inputs, DTLLM-VLT generates concise and detailed descriptions for the relevant objects. This methodology enables us to generate large-scale, diverse granularities text at low costs. The detailed workflow and ablation study has been outlined in Appendix A.3.1 and A.3.2.

Table 2: Summary of sentence number (*word number*) of four granularity generated language description in DTVLT. We using LLM and provide far more diverse semantic information based on representative environments to form our DTVLT benchmark. "Dense" indicates that provides text for every 100 frames, "initial" indicates that only the first frame of text is provided, and "concise" and "detailed" indicate the richness of information, respectively. We illustrate the diversity of text by analyzing the sentence number of texts at different granularities and the number of words.

| | Sentence Number (*Word Number*) of four granularity language description in DTVLT | | | | |
|---|---|---|---|---|---|
| | Data Source | Dense Concise | Dense Detailed | Initial Concise | Initial Detailed |
| **DTVLT** | **OTB99_Lang** | 0.6K (*3.2K*) | 0.6K (*21.9K*) | 0.1K (*0.5K*) | 0.1K (*3.6K*) |
| | **GOT-10k**[1] | 43.0K (*253.6K*) | 43.0K (*1.8M*) | 9.5K (*49.3K*) | 9.5K (*346.5K*) |
| | **LaSOT** | 35.2K (*182.6K*) | 35.2K (*1.2M*) | 1.4K (*7.1K*) | 1.4K (*47.4K*) |
| | **TNL2K** | 12.4K (*71.6K*) | 12.4K (*476.0K*) | 2.0K (*10.7K*) | 2.0K (*73.0K*) |
| | **MGIT**[2] | 16.1K (*83.2K*) | 16.1K (*553.7K*) | 0.1K (*0.6K*) | 0.1K (*4.5K*) |

[1] As the ground truth of the GOT-10k test set is not open-sourced, we only generated text for training and validation sets. And its frame rate is 10 fps, so we generate texts every 33 frames.

[2] As the ground truth of the MGIT test set is not open-sourced, we only generated text for training and validation sets.

[3] "K" stands for "thousand" and "M" stands for "million".

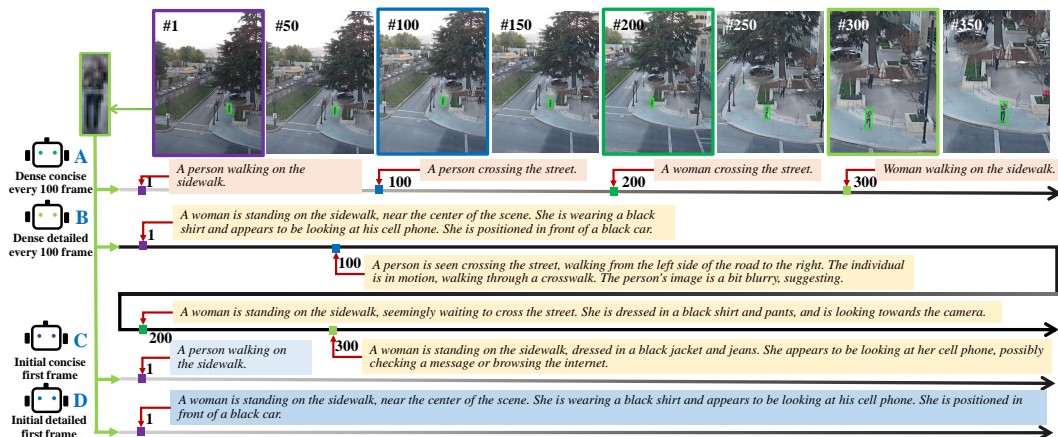

Figure 3: Examples of the text generation in DTVLT. We provide four different natural language descriptions for each video. Diverse multi-granularity text can support fine-grained evaluation of trackers, providing guidance for the development of tracking.

## 3.3 GENERATION STRATEGY

The volume and linguistic annotations of the VLT datasets determine the tracking performance. Table 1 shows that the current tracking dataset for VLT is equipped with just 5,252 (99 + 1,400 + 2,000 + 1,753) official textual descriptions. This limited data is considered inadequate for algorithms to learn effectively. These official annotations are only sufficient to describe the short-term changes of the object. The lack of textual descriptions impedes the trackers' ability to acquire a comprehensive understanding of visual contents, leading to a substantial decline in performance when generalizing to new videos. Furthermore, inaccurate textual descriptions can obstruct object tracking, turning natural language annotations into a hindrance rather than a support.

In this work, we design a multi-granular generation strategy to provide scientific natural language information. To enhance the accuracy and generality, we generate texts for five datasets to construct DTVLT, as shown in Table 2, establishing a robust foundation for VLT. This generation strategy can be expanded to more VLT and SOT datasets.

**Initial and dense text descriptions.** Inspired by the text annotations approach in OTB99_Lang (Li et al. (2017)) and TNL2K (Wang et al. (2021)), we generate text for the first frame of each video. Additionally, recognizing that 4 seconds is the threshold between human instant memory and short-term memory (Radvansky (2021); Strous et al. (1995); Atkinson & Shiffrin (1968)), we prepare for the most challenging scenario where the algorithm may not have an efficient memory mechanism. Therefore, at a frame rate of 25 FPS, equating to every 100 frames in 4 seconds, we provide the algorithm with generated text. We believe that this frequency of updates optimally maintains the algorithm's memory state and improves tracking capabilities.

**Concise and detailed text descriptions.** For the algorithm, if the BBox already adequately captures the temporal and spatial dynamics of the object, the texts should concentrate on delivering key semantic elements such as the object's category and location. When the BBox does not provide enough information for the tracker's efficient learning, more comprehensive texts are required to make up for the absent temporal and spatial connections. As a result, we generate two types of textual descriptions: concise and detailed. As depicted in Fig. 2, the concise text conveys essential information about the object, like its category (*bear*) and position (*in the water*), whereas the detailed text encompasses further spatio-temporal specifics such as color, relative position, and activities.

## 3.4 GENERATION ANALYSIS

We provide four granularities of natural language descriptions for each video, which are the initial concise description, initial detailed description, dense concise description, and dense detailed description. This is depicted in Fig. 3. For more examples of different tracking tasks in DTVLT, please refer to Appendix A.3.5. Our goal is to use diverse textual information to enhance the learning and

evaluation environment for the algorithm, as well as to offer direction for algorithmic development and model optimization.

We generate text descriptions for the DTVLT using the DTLLM-VLT (Li et al. (2024a)), which includes 26.2K initial descriptions (divided equally between 13.1K concise and 13.1K detailed texts) and 214.6K dense descriptions (also equally divided into 107.3K concise and 107.3K detailed texts). The quantity of our dense texts is 45.9 times larger than the official annotations. Additional information on the number of semantic descriptions is available in Table 1. These semantic descriptions consist of 5.2M words, featuring 17.6K non-repetitive words. Text descriptions for each frame only takes 2 seconds, and the entire method can be directly run on an RTX-3090 GPU. The vocabulary is rich, allowing for a comprehensive description of changes in the object during the tracking process. In summary, compared to previous tracking datasets, the DTVLT we constructed is a multi-task-oriented, multi-granular, large-scale dataset that utilizes LLM for automatic text annotation. Word cloud have been illustrated in Fig. 4. More detailed analysis has been outlined in Appendix A.3.3 and A.3.4.

## 4 EXPERIMENTAL RESULTS

### 4.1 DATASETS AND EVALUATION METHODS

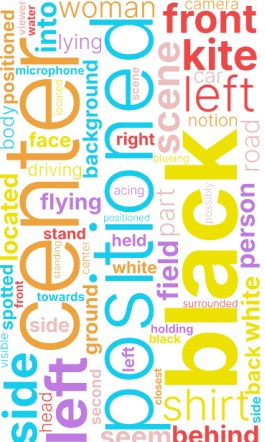

Figure 4: The word cloud of semantic descriptions.

As shown in Fig. 3, we follow generation granularities to design various mechanisms. We select several VLT trackers, MMTrack (Zheng et al. (2023)), JointNLT (Zhou et al. (2023)) and UVLTrack (Ma et al. (2024)) as baseline models and evaluate them on DTVLT (as shown in Table 3 and Fig. 5). The experimental section including both iid (independent and identically distributed) and ood (out of distribution) settings, such as LaSOT and GOT-10k being evaluated under iid and ood settings, respectively. It can verify the model's generalization ability. With the diverse environment, we can analyze the strengths and weaknesses of various trackers from the experimental results, offering insights into the development of VLT trackers. Compared with other algorithms, MMTrack is designed to be flexible with text length, avoiding the truncation of lengthy text segments. Moreover, it treats the VLT task as a token generation process, facilitating more effective learning of visual-linguistic data. While JointNLT and UVLTrack sets 50 as a maximum limit and truncates the excess information. (See Appendix B.3 for more details.)

To fairly compare the tracking performance on five datasets, we use two evaluation mechanisms. (A) We directly use the officially provided weights to test with the official annotations and on the DTVLT. (B) We retrain these models for 50 epochs on the basis of the official weights using DTVLT and test under the corresponding settings to evaluate Area Under the Curve (AUC), and tracking precision (P). For more details on the evaluation metrics, please refer to Appendix B.1.

When retraining the tracker, we selected LaSOT (Fan et al. (2019)), OTB99_Lang (Li et al. (2017)), TNL2K (Wang et al. (2021)), and RefCOCOg Mao et al. (2016b) as the training data, with a ratio of 1:1:1:1. The template image and search image sizes are 128x128 and 256x256, respectively. We used AdamW (Mao et al. (2016a)) as the optimizer and continued training for 50 epochs on the basis of the official weights, randomly sampling 30,000 image pairs per epoch. All trackers are trained on a server with four A5000 GPUs and tested on an RTX-3090 GPU. For dense text, the text is dynamically updated every hundred frames, and for initial text, only the first frame provides text information.

### 4.2 TESTING DIRECTLY ON DTVLT (MECHANISM A)

We directly use the models provided by the official for testing. From Table 3, we can draw the following conclusions: (1) Most trackers perform poorly when faced with the diverse text in DTVLT, such as JointNLT (Zhou et al. (2023)) and UVLTrack (Ma et al. (2024)), with this phenomenon being particularly prominent in JointNLT. When faced with texts not seen in the training data, JointNLT experiences a significant performance drop across various datasets. The lack of diverse VLT datasets makes it difficult for researchers to comprehensively evaluate algorithm performance when designing and testing algorithms, leading to a phenomenon similar to "memorizing the answer" observed

Table 3: Comparison with testing directly on DTVLT. The best two results are highlighted in red and blue, respectively.

| MMTrack | OTB99_Lang | | MGIT (Activity) | | LaSOT | | TNL2K | | GOT-10k | |
|---|---|---|---|---|---|---|---|---|---|---|
| | AUC | P | AUC | P | AUC | P | AUC | P | AUC | P |
| Official | 69.0 | 89.5 | 73.5 | 54.3 | 69.9 | 75.7 | 58.6 | 59.3 | - | - |
| Initial Concise | 70.6 | 91.1 | 73.9 | 54.9 | 69.0 | 74.7 | 56.6 | 56.9 | 82.9 | 79.5 |
| Initial Detailed | 68.0 | 88.4 | 72.7 | 53.4 | 68.7 | 74.4 | 55.9 | 55.4 | 82.7 | 79.0 |
| Dense Concise | 70.2 | 90.8 | 74.2 | 55.0 | 69.1 | 74.8 | 56.5 | 56.7 | 82.8 | 79.3 |
| Dense Detailed | 68.6 | 89.4 | 72.9 | 53.5 | 69.0 | 74.7 | 56.1 | 55.6 | 82.8 | 79.2 |

| JointNLT | OTB99_Lang | | MGIT (Activity) | | LaSOT | | TNL2K | | GOT-10k | |
|---|---|---|---|---|---|---|---|---|---|---|
| | AUC | P | AUC | P | AUC | P | AUC | P | AUC | P |
| Official | 65.1 | 85.3 | 58.7 | 41.3 | 60.4 | 63.6 | 57.0 | 58.2 | - | - |
| Initial Concise | 59.6 | 80.5 | 53.3 | 35.1 | 58.4 | 60.0 | 50.3 | 49.3 | 70.7 | 55.4 |
| Initial Detailed | 55.1 | 74.2 | 56.3 | 36.4 | 52.4 | 51.7 | 49.4 | 47.2 | 69.5 | 55.4 |
| Dense Concise | 58.8 | 77.9 | 48.6 | 31.1 | 58.2 | 59.4 | 50.3 | 49.3 | 70.1 | 55.1 |
| Dense Detailed | 55.1 | 74.4 | 48.9 | 28.8 | 55.6 | 54.9 | 50.0 | 48.4 | 69.2 | 54.6 |

| UVLTrack | OTB99_Lang | | MGIT (Activity) | | LaSOT | | TNL2K | | GOT-10k | |
|---|---|---|---|---|---|---|---|---|---|---|
| | AUC | P | AUC | P | AUC | P | AUC | P | AUC | P |
| Official | 68.7 | 89.0 | 64.0 | 52.2 | 67.7 | 73.7 | 62.1 | 65.6 | - | - |
| Initial Concise | 68.5 | 89.0 | 51.7 | 47.8 | 66.9 | 72.1 | 60.7 | 63.5 | 82.0 | 75.7 |
| Initial Detailed | 65.7 | 86.0 | 60.6 | 46.3 | 65.8 | 71.0 | 59.8 | 62.5 | 80.6 | 73.8 |
| Dense Concise | 67.9 | 88.1 | 60.8 | 47.1 | 67.1 | 72.4 | 60.8 | 63.6 | 82.1 | 75.8 |
| Dense Detailed | 66.1 | 86.2 | 60.7 | 46.0 | 64.1 | 71.2 | 59.8 | 62.4 | 80.7 | 73.7 |

with JointNLT and UVLTrack. (2) The approach of sequence generation is more conducive to learning unified visual-language features. It can be observed that MMTrack (Zheng et al. (2023)) has achieved further performance improvements on some datasets by incorporating the diverse texts from DTVLT, showing a stronger adaptability to text. (3) We think that the current algorithm's handling of long texts and the alignment of multiple modalities needs refinement, as it does not make the most of temporal and spatial relationships. Such temporal and spatial data are essential for enhancing tracking capabilities. In instances where the BBox's temporal-spatial details are insufficient to reliably pinpoint the object, detailed textual information is required to supply extra high-level semantic insights necessary for object tracking.

Through direct testing and comparison of tracking performance under different texts, it has been observed that the variation in texts has a significant impact on tracking performance. DTVLT compensates for the shortcomings of existing VLT datasets that cannot provide a flexible and comprehensive assessment.

## 4.3 RETRAINING AND TESTING ON DTVLT (MECHANISM B)

As previously discussed, when the dataset text information becomes denser and more accurate, it can make up for the deficiencies in the BBox annotations. The algorithm can acquire supplementary knowledge through these textual updates, which may lead to an enhancement in its performance. Consequently, we have retrained and evaluated models using a range of differently generated textual inputs. As shown in Fig. 5, we plot the performance differences between the model after retraining and direct testing, where the red line represents the mean of these differences. For the absolute performance values of the model, please refer to the detailed data in the Appendix B.2.2. By comparing with other trackers, MMTrack (Zheng et al. (2023)) and UVLTrack (Ma et al. (2024)) have seen further improvements in performance, but the performance of JointNLT (Zhou et al. (2023)) has continued to decline instead. While our hope is that the inclusion of additional modalities will

enhance the tracking performance, the current VLT trackers struggle to effectively integrate various types of information, leading to an underutilization of the multi-modal data. This outcome—where contemporary VLT trackers perform more poorly than those relying solely on visual cues—is also documented in other studies, highlighting the significant room for improvement in multi-modal tracking.

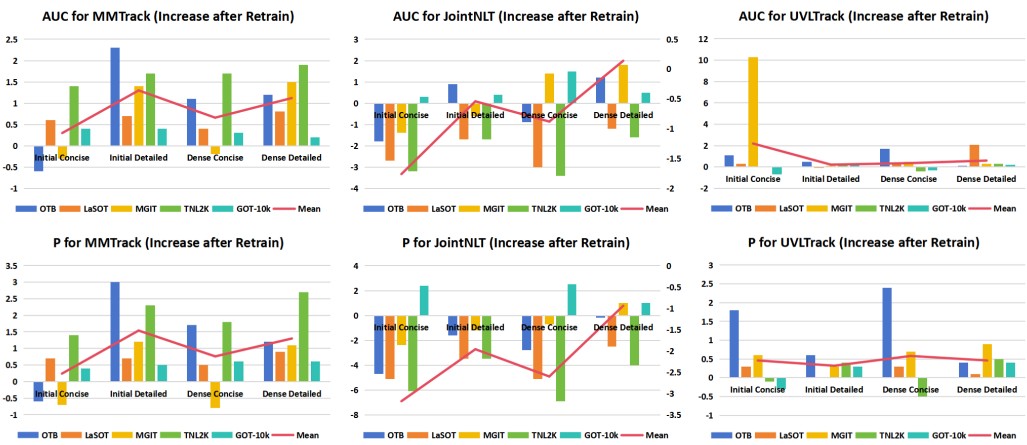

Figure 5: Comparison with retraining for 50 epochs and testing on DTVLT. We plot the performance differences between the model after retraining and direct testing, where the red line represents the mean of these differences.

By comparing results under mechanisms A and B, it is evident that in this flexible and comprehensive setting, trackers that are thoughtfully crafted (that is, those equipped with the capacity for extended input processing and the ability to align multi-modal data) can achieve superior results through diverse descriptions, rather than relying solely on brief descriptions. The experiments conducted highlight two pivotal insights: (1) Richer semantic data can enhance tracking capabilities compared to a simple sentence, which also substantiates the precision and relevance of the proposed multi-scale semantic generation strategy. (2) Providing only a basic description to VLT trackers is not practical. As a result, initiating the tracking procedure with longer and detailed sentences, or regularly updating the semantic data throughout the sequence, has proven to be more efficacious in precisely locating targets amidst intricate scenes.

## 4.4 SUMMARY

Among the three algorithms, MMTrack (Zheng et al. (2023)) demonstrated the best performance, showing some content worth further research and exploration in Mechanism A and Mechanism B. For instance, on the MGIT (Hu et al. (2023a)) dataset, dense concise text achieved optimal performance under direct testing conditions, which is somewhat different from the motivation proposed by the MGIT dataset. We believe that the current algorithms lack in long text processing and multi-modal alignment capabilities, so when facing long videos and high-difficulty sequences like MGIT, they cannot make good use of the official long text annotations. Additionally, on the OTB99_Lang (Li et al. (2017)) dataset, using initial concise text for direct testing yielded the best performance. The early datasets represented by OTB99_Lang have provided sufficient information for tracking in the BBox, and in this case, the text only needs to provide the most basic information to assist in enhancing tracking performance. This trend was further reflected after retraining and testing. We believe that MMTrack's good performance lies in its modeling approach to the VLT task, learning visual-linguistic features through sequence generation, which can enhance the generalization of the tracker.

JointNLT (Zhou et al. (2023)), as a representative of the recent SOTA algorithms, has shown surprisingly disappointing results. Both in direct testing and after retraining and testing, JointNLT's performance has declined significantly, which also confirms our analysis of the current VLT benchmark. That is, the existing text is not sufficient to support tracking like JointNLT to learn strong

visual-linguistic tracking capabilities. They still adopt a strategy of "memorizing the answer" to complete the VLT task. UVLTrack's (Ma et al. (2024)) performance falls between the two, but it also exhibits phenomena similar to JointNLT.

In summary, the emergence of DTVLT can provide a high-quality flexible experimental environment for research, and help the algorithms quickly identify bottleneck issues under various evaluation mechanisms, thereby accelerating the development of VLT algorithms.

### 4.5 VISUALIZATION AND BAD CASE ANALYSIS

We delve deeper into the limitations of the VLT algorithms through the bad cases shown in Fig. 6. The first two cases are sourced from LaSOT (Fan et al. (2019)), while the final case is taken from MGIT (Hu et al. (2023a)). MGIT and LaSOT face similar challenges in VLT task, such as interference between the object and the background, as well as significant variations in the object's appearance from the initial frame to subsequent frames. These challenges not only increase the difficulty of tracking but also affect the overall performance of existing trackers. In this context, the introduction of a diverse text generation method has become a more viable solution. By providing multi-granularity text, it enables trackers to better handle changes in the object's appearance and interference from complex backgrounds. To further enhance performance, current trackers require a more sophisticated and intelligent semantic information processing module.

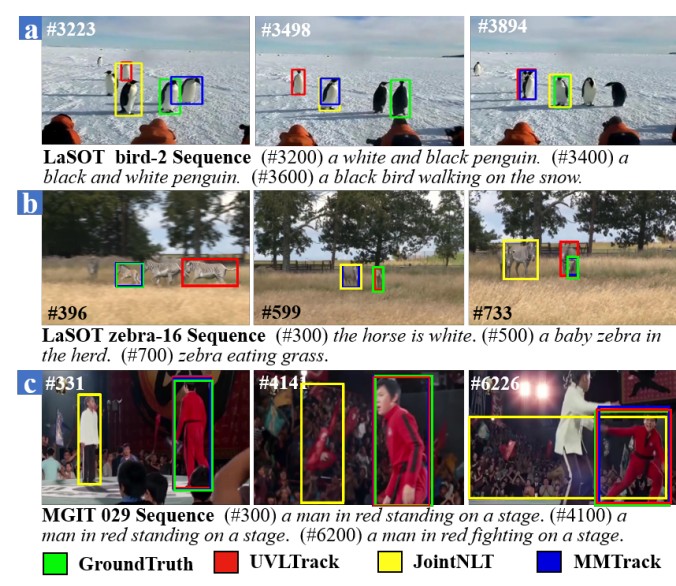

Figure 6: Visualization of tracking results on dense concise text annotations retrained algorithm. (A-B) In the LaSOT dataset, VLT trackers tend to identify and follow similar objects. (C) VLT trackers struggle to adjust to the intricacies of complex scenes, as observed in the MGIT dataset.

This module should be able to precisely extract relevant details indicated by semantic tags, helping the tracker locate and follow the object more accurately in complex scenarios. However, the current design of trackers has not yet been specifically optimized to meet this requirement, lacking mechanisms to fully utilize semantic information, which leaves room for improvement in handling complex scenarios. For more bad cases, please refer to Appendix B.4.

## 5 CONCLUSIONS

**Summary.** Object tracking forms the foundation for advanced tasks such as video understanding, and VLT may offer a promising approach to enhancing tracking capabilities. Unlike existing VLT benchmarks that primarily feature ambiguous descriptions, we (1) introduce a new VLT benchmark named **DTVLT** based on five benchmarks, and (2) develop a **multi-granular text generation strategy** to create diverse semantic information. DTVLT is the first comprehensive VLT benchmark using LLM to provide multi-granularity diverse semantic information. In conclusion, we hope this work aids researchers in advancing their studies in VLT and video understanding.

**Limitations.** Future work can address some current limitations. First, DTVLT can be expanded with additional SOT and VLT benchmarks, creating a more complex and challenging environment for tracking algorithms. Additionally, a more comprehensive evaluation system can be designed to better assess VLT and video understanding capabilities.

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

# A DATASET INFORMATION

## A.1 BASIC INFORMATION

In this work, we propose a new **v**isual **l**anguage **t**racking benchmark with **d**iverse **t**exts, named **DTVLT**, based on five prominent VLT and SOT benchmarks, including three sub-tasks: short-term tracking, long-term tracking, and global instance tracking, aiming to support further research in VLT and video understanding.

Currently, the vast majority of VLT benchmarks are annotated with a single granularity in natural language, and there is an issue of only describing the changes in the target of the first frame, which hinders the algorithm's understanding of the video content. The algorithms tend to adopt a 'memorizing the answer' approach to accomplish the task of object tracking. This phenomenon highlights the constraints imposed by single granularity text descriptions on the comprehension of long videos. Consequently, our research endeavors to incorporate diverse semantic cues, with the goal of equipping algorithms to more effectively navigate the intricate narrative dynamics inherent to target tracking and the understanding of video contents.

## A.2 DATA SELECTION

We have selected five representative benchmarks, covering short-term tracking (OTB99_Lang (Li et al. (2017)), GOT-10k (Huang et al. (2021)) and TNL2K (Wang et al. (2021))), long-term tracking (LaSOT (Fan et al. (2019))), and global instance tracking (MGIT (Hu et al. (2023a))) tasks. The number of videos in each dataset and the number of texts officially annotated are shown in Table A1.

Table A1: Summary of selected datasets.

| Dataset | Number of Videos | | Official Annotations |
|---|---|---|---|
| | Train | Evaluation | |
| OTB99_Lang (Li et al. (2017)) | 51 | 48 | 99 |
| GOT-10k (Huang et al. (2021)) | 9,335 | 360 | 0 |
| LaSOT (Fan et al. (2019)) | 1,120 | 280 | 1,400 |
| TNL2K (Wang et al. (2021)) | 1,300 | 700 | 2,000 |
| MGIT (Hu et al. (2023a)) | 105 | 45 | 1,753 |

## A.3 DIVERSE TEXT GENERATION

### A.3.1 GENERATION PIPELINE

The pipeline of text generation in DTVLT with DTLLM-VLT (Li et al. (2024a)) is depicted in Fig. A1. An input video frame accompanied by the respective object BBox is processed by SAM (Kirillov et al. (2023)), which employs an image encoder, a prompt encoder, and a mask decoder to extract the masks of the object in question. These masks, along with the video frame, are then fed into Osprey (Yuan et al. (2023)). Within Osprey, the images and masks undergo encoding, are merged with pre-defined prompts, and subsequently, the system leverages a LLM (Chiang et al. (2023); Touvron et al. (2023)) to produce succinct and comprehensive descriptions for the objects. This methodology allows for the generation of large-scale, diverse granularities textual data for SOT and VLT datasets at minimal expense.

### A.3.2 REASON FOR SELECTING LLM AND ABLATION STUDY

We will introduce the reason for selecting LLM and corresponding ablation study.

**Purpose of using LLM for DTVLT**: Existing benchmarks provide video-level text that struggles to effectively capture the dynamic changes in video content, which also hinders the development of efficient video-language trackers. Therefore, providing more diverse semantic descriptions that align with the dynamic characteristics of videos for existing VLT benchmarks can offer a rich data environment for further algorithm optimization and holds significant research value. To achieve this

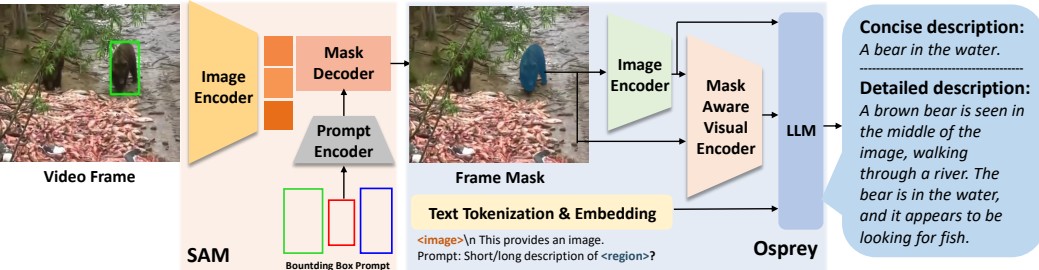

Figure A1: The pipeline of text generation for DTVLT based on DTLLM-VLT (Li et al. (2024a)), which can provide dense categories on MGIToncise/detailed text generation based on given video frames and BBox of object.

Table A2: Ablation Study of SAM (Kirillov et al. (2023)) and LLM Backbone with Text Similarity Comparison on OTB99_Lang (Li et al. (2017)). For SAM, we replaced the SAM-B with SAM-H and SAM-L, and for Osprey (Yuan et al. (2023)), we replaced the Osprey with Osprey-chat and Controlcap (Zhao et al. (2024)). We report BLEU (Papineni et al. (2002)), GLEU (Mutton et al. (2007)), METEOR (Banerjee & Lavie (2005)), Recall, Precision and F1 score (Goutte & Gaussier (2005)). We compared the similarity between the texts generated after replacing the backbone and the DTVLT texts, so the scores of ours (combining SAM-B with Osprey) are all 1.00.

| Text Similarity Comparison on OTB99_Lang | | | | | | | | |
| --- | --- | --- | --- | --- | --- | --- | --- | --- |
| SAM Backbone | LLM Backbone | Text Granularity | BLEU | GLEU | METEOR | Recall | Precision | F1 |
| **SAM-H** | **Osprey** | Dense Concise | 0.76 | 0.79 | 0.84 | 0.87 | 0.87 | 0.87 |
| | | Dense Detailed | 0.56 | 0.57 | 0.68 | 0.72 | 0.71 | 0.71 |
| **SAM-L** | | Dense Concise | 0.68 | 0.72 | 0.79 | 0.85 | 0.87 | 0.86 |
| | | Dense Detailed | 0.56 | 0.58 | 0.67 | 0.74 | 0.73 | 0.72 |
| **SAM-B** | **Osprey-Chat** | Dense Concise | 0.25 | 0.34 | 0.48 | 0.59 | 0.60 | 0.60 |
| | | Dense Detailed | 0.17 | 0.23 | 0.41 | 0.48 | 0.51 | 0.49 |
| | **ControlCap**[1] | Dense Concise | 0.09 | 0.14 | 0.28 | 0.30 | 0.37 | 0.34 |

[1] ControlCap cannot generate detailed text, we only analyze its concise text.

goal, we need to using LLM and construct a pipeline to understand the dynamic changes in the video process, especially to effectively perceive fine-grained target variations.

**Reason for choosing SAM and Osprey**: Currently, the BBox in tracking benchmarks only provides relatively coarse information. Thus, we first obtain masks through SAM (Kirillov et al. (2023)) and acquire pixel-level information of the object, laying a solid foundation for fine-grained perception. On this basis, Osprey's (Yuan et al. (2023)) goal is to achieve pixel-level understanding, which coincides with our needs for dynamic changes in the tracking process. In addition to inputting complete images for perceiving foreground and background information, it also provides a fine-grained encoder for masks, which will enhance the understanding of dynamic environments. Moreover, Osprey's training data is regenerated through GPT Achiam et al. (2023) and is currently the only model that can provide detailed text descriptions for tracking objects in region-level caption field. The entire method is plug-and-play, and each module can be replaced at any time.

**Details of ablation study**: We conduct a detailed ablation analysis of SAM (Kirillov et al. (2023)) and Osprey (Yuan et al. (2023)). We employed two types of metrics to compare the results of the ablation study, including Precision and AUC for tracking, and Recall, Precision, F1 (Goutte & Gaussier (2005)), BLEU Papineni et al. (2002), GLEU Mutton et al. (2007) and METEOR Banerjee & Lavie (2005) for text similarity comparison, to evaluate the choice of the backbone from multiple perspectives. For SAM, we replaced the SAM-B with SAM-H and SAM-L, and for Osprey, we replaced the Osprey with Osprey-chat and Controlcap (Zhao et al. (2024)). Since ControlCap cannot generate detailed text, we only analyzed its concise text. When comparing text similarity, we compared the similarity between the texts generated after replacing the backbone and the DTVLT texts.

Table A3: Ablation Study of SAM (Kirillov et al. (2023)) and LLM Backbone with Visual Language Tracking Performance on OTB99_Lang (Li et al. (2017)). For SAM, we replaced the SAM-B with SAM-H and SAM-L, and for Osprey (Yuan et al. (2023)), we replaced the Osprey with Osprey-chat and Controlcap (Zhao et al. (2024)). We test directly on DTVLT (Mechanism A) and report AUC and Precision score.

| Visual Language Tracking Performance on OTB99_Lang | | | | |
|---|---|---|---|---|
| SAM Backbone | LLM Backbone | Text Granularity | Precision | AUC |
| **SAM-B (Ours)** | **Osprey (Ours)** | Official | 89.5 | 69.0 |
| | | Dense Concise | 90.8 | 70.2 |
| | | Dense Detailed | 89.4 | 68.6 |
| **SAM-H** | **Osprey** | Dense Concise | 90.5 | 69.9 |
| | | Dense Detailed | 90.6 | 69.6 |
| **SAM-L** | | Dense Concise | 90.0 | 69.7 |
| | | Dense Detailed | 90.4 | 69.4 |
| **SAM-B** | **Osprey-Chat** | Dense Concise | 88.8 | 68.5 |
| | | Dense Detailed | 89.0 | 68.2 |
| | **ControlCap**[1] | Dense Concise | 90.7 | 69.9 |

[1] ControlCap cannot generate detailed text, we only analyze its concise text.

**Conclusion of ablation study**: We present the results of the ablation study. From Table A2, it can be observed that replacing SAM (Kirillov et al. (2023)) does not significantly affect text generation; the text similarity metrics are all very high. Considering both the time cost and performance, we have chosen SAM-B. However, after replacing the LLM backbone, there is a noticeable decrease in the metrics. This is because different LLMs have different training data and strategies, and the descriptions for the same object may vary in style. Our goal is to use diverse texts for tracking tasks, so we further conducted ablation analysis for tracking experiments based on the generated texts. In Table A3, we conducted tracking experiments on the OTB99_Lang (Li et al. (2017)) dataset for dense texts. The backbone model we used achieved the best performance on OTB99_Lang, which further verifies the importance and rationality of the texts generated by DTVLT. And it can be seen that our proposed framework still enhances tracking performance with most dense texts after replacing different modules.

### A.3.3 GENERATION ANALYSIS

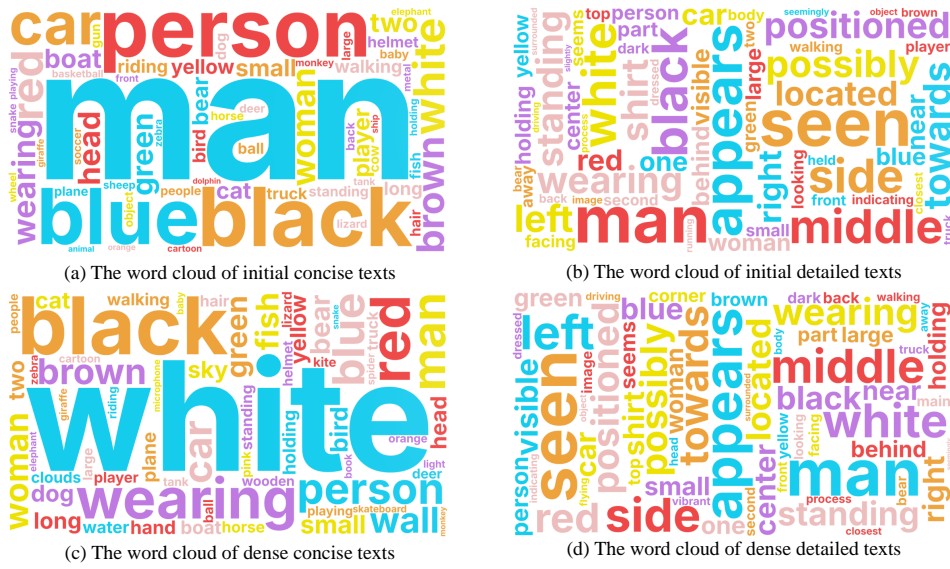

(a) The word cloud of initial concise texts

(b) The word cloud of initial detailed texts

(c) The word cloud of dense concise texts

(d) The word cloud of dense detailed texts

Figure A2: word cloud of four granularities on DTVLT.

We calculated the visual-textual similarity between the official annotations and the text generated by DTVLT with respect to the tracking targets. Specifically, we cropped the target from the video frames according to the BBox, and then calculated the similarity between the text and the target using CLIP (Radford et al. (2021)). The visual-textual similarity between the official text annotations and the target is 0.187. In the DTVLT, the visual-textual similarity for the concise text and detailed text generated with respect to the target are 0.185 and 0.195, respectively. We think the visual-textual similarity of the text generated in DTVLT is comparable to that of the manually annotated text. The word cloud of various texts is shown in Fig. A2. Our generated texts are more diverse than official annotations.

### A.3.4 GENERATION FILE

We propose a **multi-granular generation strategy** to generate the semantic description, and use txt format to save the natural language annotation for each video sequence. Here we illustrate an example to show the txt file structure for video sequence *bear-17* with dense detailed texts in the LaSOT benchmark, as shown in Listing 1 and Fig. A3. Due to the limited space, we only illustrate some representative information, with additional information of a similar structure represented by ellipses. Please download and check the dataset for more detailed annotation about each video sequence.

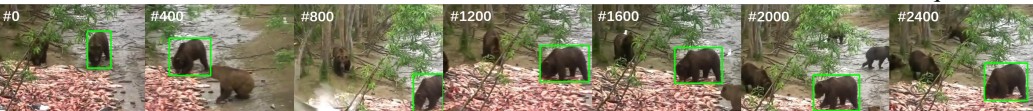

**Official**: *brown bear hunting on the ground*
**Initial Concise**: *A bear in the water.*
**Initial Detailed**: *A bear is seen in the right part of the image, walking through the water. It appears to be looking for fish to eat.*
**Dense Concise**: (#0) *a bear in the water.* (#100) *a bear in the water.* (#400) *a bear in the water.* (#800) *a bear eating fish.* (#1200) *a bear eating fish.* (#1600) *a bear eating salmon.* (#2000) *a bear eating salmon.* (#2400) *a bear eating fish.*
**Dense Detailed**: (#0) *A bear is seen in the right part of the image, walking through the water. It appears to be looking for fish to eat.* (#100) *A bear is seen in the right part of the image, wading through the river with its front paws in the water. It appears to be looking for fish to eat.* (#400) *A darker brown bear is positioned on the left side of the image, and it appears to be behind another bear. This bear is partially obscured, with only its head and shoulders visible.* (#800) *On the right side of the image, a brown bear is seen in the water, holding a fish in its mouth. The bear appears to be enjoying its meal, and it's the only one in the scene to be fully in the water.* (#1200) *On the right side of the image, a bear is seen in the background, partially hidden behind a tree. This bear appears to be looking for fish in the river, but its face is not visible.* (#1600) *A bear is seen in the background, partially hidden behind some trees. It appears to be looking for fish in the river, with its head turned to the side.* (#2000) *A bear is seen in the front right of the image, its face is fully visible. It appears to be a young bear, possibly a cub, and it is actively eating salmon.* (#2400) *On the right side of the image, a bear is seen with its back towards the camera. It appears to be wading through a river, and its fur is darker compared to the other bear.*

Figure A3: Example of texts generation for DTVLT.

1. For each sequence, we save the following information in the txt file:

   (a) *frame_id*: The frame id of the sequence. Note that in the txt file, we use 0 to represent the first frame.
   (b) *description*: The natural language description generated by DTLLM-VLT.

### A.3.5 MORE EXAMPLE IN DTVLT

DTVLT encompasses three tracking tasks designed to assess the capabilities of tracking systems under varying conditions and durations. The three tasks are:

**Short-term Tracking (STT).** This task focuses on tracking objects over short sequences where the object remains visible throughout. It tests a tracker's ability to handle rapid movements, occlusions, and changing environments over brief periods.

**Long-term Tracking (LTT).** LTT challenges trackers to maintain the identity of objects over extended sequences, where the object might disappear and reappear. It evaluates the endurance of trackers in maintaining tracking consistency over long durations and their ability to re-identify the object after loss of track.

**Global Instance Tracking (GIT).** This involves tracking an object across different scenes and conditions, possibly even when the object changes its appearance significantly. GIT tests a tracker's ability to generalize the object's identity across various scenarios and to handle large-scale variations in appearance.

The DTVLT framework provides a multi-granular textual description for these tasks, enabling a detailed evaluation of each tracker's performance on specific challenges posed by different tracking

**Listing 1** The txt file about the dense detailed information of video sequence *bear-17* in LaSOT (Fan et al. (2019))

```
{
    "0": "A bear is seen in the right part of the image, walking
         through the water. It appears to be looking for fish to
         eat.",
    "100": "A bear is seen in the right part of the image, wading
         through the river with its front paws in the water. It
         appears to be looking for fish to eat.",
    "200": "A bear is seen in the right part of the image, with
         its back towards the camera. It appears to be wading
         through a river, with its head submerged in the water. The
         bear seems to be enjoying its time in the water.",
    "300": "...",
    "...": "...",
    "2300": "...",
    "2400": "..."
}
```

environments. Such a setup not only pinpoints the strengths and weaknesses of each tracking algorithm but also offers insights that can drive innovations in tracker design. Examples of each tracking task in DTVLT are illustrated in Fig. A7 (STT), Fig. A5 (LTT) and Fig. A6 (GIT).

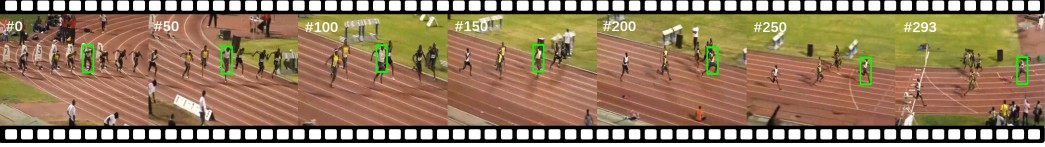

**Short-term Tracking — OTB99_Lang**
**Initial Concise**: *A man running on a track.*
**Initial Detailed**: *The third runner from the left, who is positioned at the center of the image, is a notable participant in the race. He is in the middle of the group and is not the last runner.*
**Dense Concise**: (#0) *A man running on a track.* (#100) *A man wearing a white shirt.* (#200) *A man running on a track.*
**Dense Detailed**: (#0) *The third runner from the left, who is positioned at the center of the image, is a notable participant in the race. He is in the middle of the group and is not the last runner.* (#100) *A man in the center of the image, wearing a white shirt with the number 6 on it, is running. He is the third runner from the left and is positioned between two other runners.* (#200) *A man, dressed in black shorts, is seen running on the track. He is located towards the right side of the image, and appears to be in motion, possibly participating in the race.*

Figure A4: Example for Short-term Tracking (STT) in DTVLT.

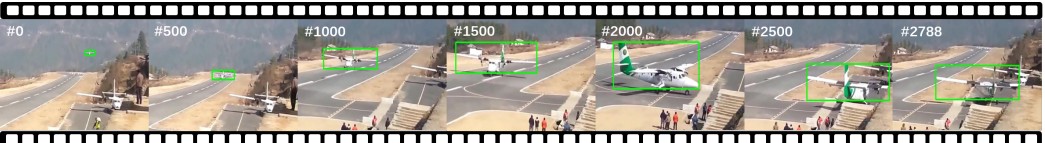

**Long-term Tracking — LaSOT**
**Initial Concise**: *A plane in the sky.*
**Initial Detailed**: *A small airplane is seen flying in the air, possibly coming in for a landing. The plane is located towards the top right of the image, and it appears to be a smaller aircraft compared to the other plane in the scene.*
**Dense Concise**: (#0) *A plane in the sky.* (#500) *A white airplane on the runway.* (#1000) *The plane is white.* (#1500) *The plane is on the runway.* (#2000) *The plane is white.* (#2500) *A green and white airplane.* (#2700) *A long white wing.*
**Dense Detailed**: (#0) *A small airplane is seen flying in the air, possibly coming in for a landing. The plane is located towards the top right of the image, and it appears to be a smaller aircraft compared to the other plane in the scene.* (#500) *A small airplane is seen in the process of taking off from the runway. The aircraft is positioned on the right side of the image, and its front is pointing upwards, indicating the beginning of its ascent into the sky.* (#1000) *A large airplane is seen on the runway, positioned to take off. The plane is mostly white, with a distinct green tail. It's located towards the left side of the image, and the front of the plane is facing towards the viewer.* (#1500) *A large airplane is seen in the process of taking off from the runway. The plane is mostly white, with a distinct green and yellow flag on its tail. It's positioned towards the left side of the image, and the scene is set in a grassy field.* (#2000) A large white airplane is seen in the process of taking off from the runway. The plane's front wheels are off the ground, indicating that it is about to become fully airborne. The scene is set in a sunny location, with the airplane's green and white colors standing out against the backdrop.* (#2500) *A large green and white airplane is parked on the runway. The plane's tail is facing towards the camera, and it's positioned in such a way that the wing is pointing towards the viewer.* (#2700) *A large airplane is parked on the runway, with its tail pointing upwards. The plane's wings span out widely, creating a significant shadow on the ground. The plane's body is white, contrasting with its darker tail.*

Figure A5: Example for Long-term Tracking (LTT) in DTVLT.

**Global Instance Tracking — MGIT**
**Initial Concise**: *A white bird with an orange beak.*
**Initial Detailed**: *A white bird, possibly a goose or a swan, is standing on the grass. It is located towards the left side of the image and appears to be looking away from the camera.*
**Dense Concise**: *(#0) A white bird with an orange beak. (#1000) A white bird standing on a porch. (#5000) A white and orange bird. (#6000) A swan in a blue pool. (#7000) A swan in a pool. (#11000) A white swan in the water.. (#12700) A white swan in the water..*
**Dense Detailed**: *(#0) A white bird, possibly a goose or a swan, is standing on the grass. It is located towards the left side of the image and appears to be looking away from the camera. (#1000) A white bird, possibly a goose or a heron, is standing on the wooden deck of a house. It appears to be looking down, perhaps at the ground or something on the deck. (#5000) A large, white, and yellow bird, possibly a swan or a goose, is standing on the grass. It is facing towards the right and its beak is open, giving it an interesting and somewhat comical appearance. (#6000) A large white swan is standing in a blue water feature, possibly a bath or a pond. The bird is drinking water from the pond, and it seems to be enjoying the cool water. (#7000) A white swan is standing on the right side of the image. It's in a blue water bowl and seems to be drinking. The swan is mostly white, with a few brown markings on its head. (#11000) A white swan is standing in the water, its neck is straight and it appears to be looking upwards. It's positioned towards the right side of the image. (#12700) A white swan is standing in the water near the edge of a pond, close to a bank. It appears to be looking for food. The swan is near a ramp and a bridge, and it's positioned in front of a body of water.*

Figure A6: Example for Global Instance Tracking (GIT) in DTVLT.

# B EXPERIMENT INFORMATION

## B.1 EVALUATION METRICS

Precision P typically refers to the proportion of frames where the distance between the tracker's predicted bounding box and the ground-truth bounding box is less than or equal to a given threshold $\theta_d$. The specific definition is as follows:

$$\mathrm{P}(E) = \frac{1}{|E|} \sum_{l=1}^{|E|} \frac{1}{|L|} |\{t : d_t \leq \theta_d\}| \tag{1}$$

Here, $|E|$ represents the total number of sequences in the dataset, $|L|$ represents the number of frames in sequence $l$, $d_t$ is the distance between the predicted position $p_t$ and the ground-truth position $g_t$ in frame $t$, and $\theta_d$ is a preset threshold.

AUC is the area under the cumulative distribution function (CDF) of Precision P calculated at different $\theta_d$ values. It provides a comprehensive measure to evaluate the performance of the tracker at various distance thresholds. The value of AUC ranges between 0 and 1, with higher values indicating better tracker performance. The definition of AUC is as follows:

$$\mathrm{AUC} = \int_0^{d_{\max}} \mathrm{P}(\theta_d) \, d\theta_d \tag{2}$$

Here, $d_{\max}$ is the maximum possible distance value, and $\mathrm{P}(\theta_d)$ is the precision at a specific threshold $\theta_d$. The calculation of AUC is typically done by plotting the Precision-Recall Curve at a range of $\theta_d$ values, then approximating the integral using numerical integration methods (such as the trapezoidal rule).

In practice, AUC is obtained by plotting the Precision-Recall Curve, where Precision P is the point on the curve, and Recall is the proportion of frames correctly predicted by the tracker to the total number of frames. The AUC is the area under this curve.

## B.2 EVALUATION MECHANISM

### B.2.1 MECHANISM A

To evaluate the performance of existing algorithms on DTVLT, we implement Mechanism A. Utilizing the official weight files provided (URLs as shown in Table A4), we keep all parameters unchanged. During the evaluation process, we replace the official texts with texts from DTVLT to test the

performance of various VLT algorithms under the initialization conditions of Natural Language (NL) and Bounding Box (BBox).

### B.2.2 MECHANISM B

Furthermore, we retrain three algorithms and then retest them on DTVLT, establishing Mechanism B. Specifically, we continue training for an additional 50 epochs based on the official weights, using datasets such as OTB99_Lang (Li et al. (2017)), LaSOT (Fan et al. (2019)), TNL2K (Wang et al. (2021)), and RefCOCOg (Mao et al. (2016b)). During the training process, we replace the official texts with different texts. After the training was completed, we reassess the performance of the algorithms under the corresponding text settings.

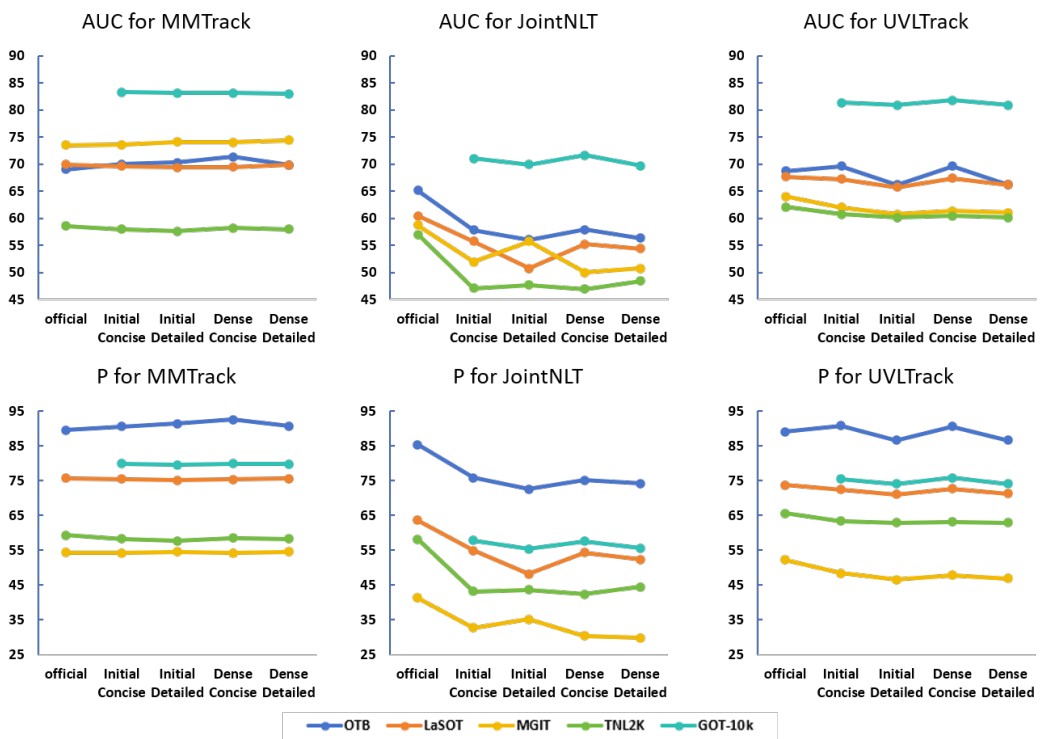

Figure A7: Comparison with retraining for 50 epochs and testing on DTVLT.

### B.3 BASELINE INFORMATION

Detailed information about the baselines are illustrated in Table A4, we use the parameters provided by the original authors.

Table A4: Table: The model architectures and URLs of open-sourced algorithms used in this work.

| Tracker | Architecture | Initializa | URL |
|---|---|---|---|
| JointNLT (Zhou et al. (2023)) | Transformer | NL & BBox | https://github.com/lizhou-cs/JointNLT |
| MMTrack (Zheng et al. (2023)) | Transformer | NL & BBox | https://github.com/Azong-HQU/MMTrack |
| UVLTrack (Zheng et al. (2023)) | Transformer | NL & BBox | https://github.com/OpenSpaceAI/UVLTrack |

JointNLT (Zhou et al. (2023)) operates as a combined visual grounding and tracking framework, utilizing natural language specifications for tracking. This framework integrates the tasks of tracking and grounding, allowing it to handle various references within these processes. Moreover, it introduces a semantics-guided temporal modeling module, which offers temporal cues derived from historical predictions to the joint model, thereby enhancing its ability to adapt to changes in the appearance of the object.

MMTrack (Zheng et al. (2023)) redefines vision-language tracking by conceptualizing it as a token generation task. It develops an innovative pipeline that taps into the capabilities of VL multi-modal learning from a holistic modeling standpoint. The method is both simple and adaptable, combining language and bounding boxes into multi-cue token inputs. It simplifies the process by discarding unnecessary sub-task learning and optimization goals, using cross-entropy solely as its single training objective.

UVLTrack (Ma et al. (2024)) presents a groundbreaking unified tracker for both visual and vision-language tracking, which is adept at managing three distinct types of target references (BBOX, NL, NL+BBOX) simultaneously. It has engineered a modality-unified feature extractor that facilitates the concurrent learning of visual and language features and implements a multi-modal contrastive loss to integrate these modal features into a cohesive semantic framework. It introduces a modality-adaptive box head that effectively extracts scenario-specific features from various modal references and precisely localizes the target through a contrastive approach, thereby boosting its robust performance in all reference scenarios.

### B.4 MORE BAD CASES

In order to demonstrate how different algorithms perform in diverse environments, We specifically selected several cases where they performed poorly as shown in Fig. A8, A9 and A10. These examples show that only in a diverse environment can we observe the strengths and weaknesses of algorithms in finer detail. Therefore, providing such an environment is crucial as it enables us to more accurately assess and compare the performance of various algorithms.

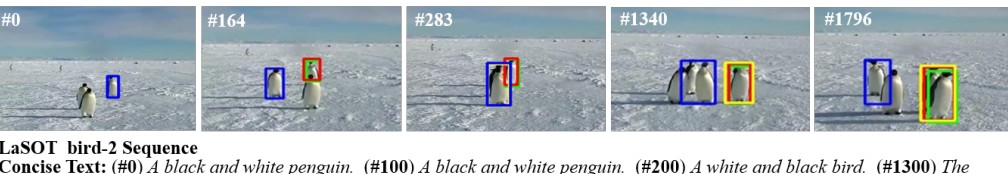

**LaSOT bird-2 Sequence**
**Concise Text:** (**#0**) *A black and white penguin.* (**#100**) *A black and white penguin.* (**#200**) *A white and black bird.* (**#1300**) *The penguin is white.* (**#1700**) *A black and white penguin.*
**Detailed Text:** (**#0**) *A lone penguin is spotted in the middle of the snowy field, facing towards the viewer. This penguin is distinct from the others as it's not in a group and is located towards the right side of the image.* (**#100**) *A lone black and white penguin is situated on the right side of the image, away from the group. This bird is facing the camera, giving us a clear view of its face.* (**#200**) A lone bird is spotted in the middle of the icey landscape. It's positioned slightly towards the right side of the image, facing towards the left. This bird is noticeably away from the group, giving it a sense of solitude.* (**#1300**) *The penguin on the far right of the image is the closest one to the viewer. It is standing on the snowy ground, facing towards the right.* (**#1700**) *The penguin on the far right of the image is the closest one to the viewer. It is standing on the snowy ground, looking straight ahead.*

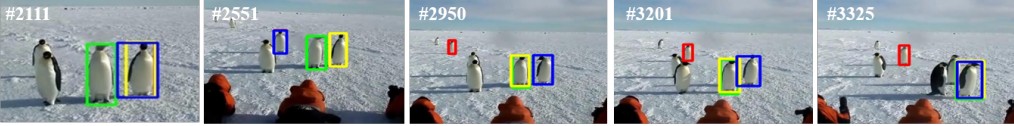

**LaSOT bird-2 Sequence**
**Concise Text:** (**#2100**) *The penguin is white.* (**#2500**) *A white and black penguin.* (**#2900**) *The penguin is white.* (**#3200**) *A white and black penguin.* (**#3300**) *A black and white penguin.*
**Detailed Text:** (**#2100**) *The middle penguin, which is the second from the left, is looking directly at the camera. This penguin is the shortest among the group and is positioned between the other two penguins.* (**#2500**) *The second penguin from the right, which is also the third penguin from the left, is standing on the snowy ground. This penguin is part of a group of five penguins that are spread out across the snowy field.* (**#2900**) *The second penguin from the left, which is also the third penguin if counting from the right. This penguin is walking in the snow, and it's not the closest to the viewer.* (**#3200**) *The second penguin from the right, which is also the third penguin if counting from the left. This penguin is standing in the snow, and it's facing towards the viewer.* (**#3300**) *A black and white penguin is seen on the right side of the image. It's the second penguin from the right and is looking directly at the camera.*

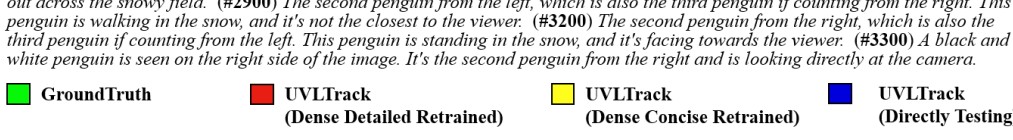

Figure A8: Bad Case for UVLTrack in DTVLT.

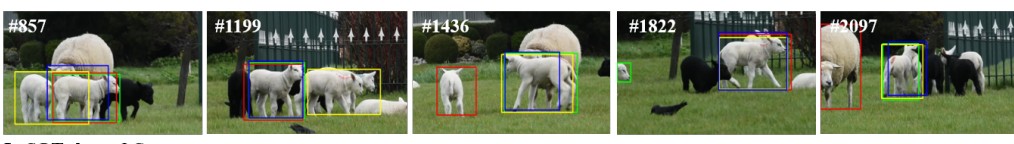

**LaSOT zebra-16 Sequence**
**Concise Text:** (**#0**) *A baby zebra walking.* (**#200**) *The zebra is eating.* (**#400**) *Zebra in the field.* (**#600**) *A person standing in the field.* (**#700**) *Zebra eating grass.*
**Detailed Text:** (**#0**) *A zebra is seen walking behind another zebra, slightly hidden. It appears to be following the other zebra, perhaps in a herd or group.* (**#200**) *A zebra is seen eating grass, standing slightly behind a wired fence. It appears to be grazing in a field, with a bush in the background.* (**#400**) *A zebra is seen in the middle of the herd, surrounded by other zebras. It appears to be a young zebra, possibly a baby, as it is smaller in size compared to the others.* (**#600**) *A cow is standing in the middle of a field, close to a fence. It is facing towards the right, and there's a man in a grey shirt standing in front of it. The cow appears to be calmly grazing on the grass.* (**#700**) *A zebra is seen standing in the middle of two other zebras, facing to the left. This zebra is located in the center of the image, and it appears to be looking at something off-camera.*

**LaSOT zebra-16 Sequence**
**Concise Text:** (**#700**) *Zebra eating grass.* (**#900**) *A baby zebra walking.* (**#1100**) *A baby zebra in the herd.* (**#1200**) *Zebra is looking at the ground.*
**Detailed Text:** (**#700**) *A zebra is seen standing in the middle of two other zebras, facing to the left. This zebra is located in the center of the image, and it appears to be looking at something off-camera.* (**#900**) *A zebra is standing in the middle of the field, facing towards the right. It is noticeable as it is the only zebra in the group that is looking in that direction.* (**#1100**) *A zebra is standing slightly behind the others, looking directly at the camera. This zebra is positioned between two other zebras, creating a sense of depth in the image.* (**#1200**) *A zebra is standing in the middle of two other zebras, facing towards the right. This zebra is distinctly positioned between its companions, with its head held high.*

■ **GroundTruth**  ■ **MMTrack**  ■ **MMTrack**  ■ **MMTrack**
           **(Dense Detailed Retrained)**  **(Dense Concise Retrained)**  **(Directly Testing)**

Figure A9: Bad Case for MMTrack in DTVLT.

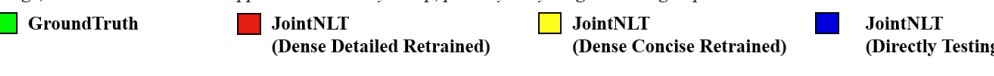

**LaSOT sheep-3 Sequence**
**Concise Text:** (**#0**) *A white lamb.* (**#100**) *A white lamb in the grass.* (**#300**) *A white lamb.* (**#500**) *A white lamb in a field.* (**#600**) *A white lamb.*
**Detailed Text:** (**#0**) *A white lamb is standing on the left side of the image. It is the only one of the three lambs that is not eating.* (**#100**) *A small lamb is positioned in the middle of two larger sheep. This lamb is the smallest of the three and is located between a larger sheep and a black sheep.* (**#300**) *A white lamb is standing in the middle of two other lambs. It is facing to the right and appears to be looking at something.* (**#500**) *A sheep is located towards the left side of the image, appearing to be the closest to the viewer. This sheep is facing away from the camera, showing its rear.* (**#600**) *On the left side of the image, a small, white lamb is seen. It appears to be a baby sheep, as it is smaller than the others and is not eating.*

**LaSOT sheep-3 Sequence**
**Concise Text:** (**#800**) *A white lamb eating grass.* (**#1100**) *A white lamb.* (**#1400**) *A white lamb.* (**#1800**) *A white lamb walking.* (**#2000**) *A white lamb in the grass.*
**Detailed Text:** (**#800**) *A white lamb is standing in the middle of two other lambs, one black and one white. It appears to be a young sheep, possibly a baby.* (**#1100**) *On the far left of the image, a sheep is seen with its back turned towards the camera. It appears to be the only one in the group not looking towards the camera.* (**#1400**) *A small lamb is located in the middle of the group, sandwiched between two larger sheep. It's the smallest of the three sheep and is facing the camera.* (**#1800**) *On the far left of the image, a small lamb is seen. It appears to be a baby sheep, possibly the youngest in the group.*

■ **GroundTruth**  ■ **JointNLT**  ■ **JointNLT**  ■ **JointNLT**
           **(Dense Detailed Retrained)**  **(Dense Concise Retrained)**  **(Directly Testing)**

Figure A10: Bad Case for JointNLT in DTVLT.