# OpenReview forum: "DTVLT: A Multi-modal Diverse Text Benchmark for Visual Language Tracking Based on LLM"
_ICLR.cc/2025/Conference — ICLR 2025 Conference Withdrawn Submission_

### Official Review · Reviewer_G4H8 · 2024-10-26

**Soundness:** 3
**Presentation:** 3
**Contribution:** 3
**Rating:** 5
**Confidence:** 3

**Summary:**

This paper proposes DTVLT, a novel benchmark for Visual Language Tracking that integrates diverse text annotations across several prominent VLT and SOT benchmarks, which is generated via the usage of LLMs. DTVLT includes four levels of text granularity, allowing for a richer representation of video content, which is beneficial for VLT and video understanding research. Authors also conduct well-rounded experiments evaluating the impact of diverse text on tracking performance.

**Strengths:**

1. The presentation is quite clear and the motivation is sound. Figures and tables are well-illustrated.
2. The introduced DTVLT is a comprehensive benchmark that incorporates diverse text annotations, addressing the limitations of existing VLT benchmarks which often rely on uniform descriptions.
3. The authors claim that the uniformity of texts may lead to "memorizing the answers”, and they conduct relevant experiments on the new benchmark to testify that variation in texts has a significant impact on tracking performance, which further bolsters the significance of DTVLT.

**Weaknesses:**

1. The data generation pipeline as well as the data collection procedure are lack of novelty, by simply using previous DTLLM-VLT and VLT and SOT benchmarks.
2. The quality of the benchmark relies heavily on the performance of DTLLM-VLT, and there seems to be no correction/filtering strategy to compensate for this.
3. The evaluation of VLT methods should be more comprehensive, including more NLT approaches to further validate the effectiveness of DTVLT

**Questions:**

See weaknesses

---

### Official Review · Reviewer_Zs9H · 2024-10-29

**Soundness:** 3
**Presentation:** 3
**Contribution:** 3
**Rating:** 5
**Confidence:** 3

**Summary:**

This paper addresses the limitations of current Visual Language Tracking (VLT) benchmarks, which often rely on simple, human-annotated text descriptions that lack nuance, stylistic variety, and granularity, leading algorithms to adopt a memorization approach rather than achieving genuine video content understanding. To overcome this, the authors leverage large language models (LLMs) to generate diverse semantic annotations for existing VLT and single object tracking (SOT) benchmarks, creating a new benchmark called DTVLT. DTVLT includes varied text descriptions across multiple levels of detail and frequency, supporting three sub-tasks: short-term tracking, long-term tracking, and global instance tracking. Through the method DTLLM-VLT, they produce high-quality, world-knowledge-rich descriptions in four levels of granularity. Experimental analyses on DTVLT reveal the effects of text diversity on tracking performance, aiming to uncover current algorithm limitations and drive advancements in VLT and video understanding research.

**Strengths:**

1.   This work presents a highly promising and comprehensive research direction for the community by providing annotations at different granularities across multiple scales, addressing a significant gap in current Visual Language Tracking (VLT) and Single Object Tracking (SOT) benchmarks. The addition of annotations at various levels of detail and frequency enriches the benchmark and enhances its adaptability to different levels of semantic complexity, which is crucial for future VLT research. By covering short-term, long-term, and global instance tracking sub-tasks, this benchmark enables a broader application range and contributes to a nuanced understanding of tracking performance across varying degrees of difficulty, making it a valuable resource for the community.
2.    The paper is very well-written, with a clear and concise structure that effectively conveys the research objectives, methodology, and findings. The coherent organization and compact style allow readers to easily follow the research narrative and understand the underlying significance of the proposed benchmark. The structured approach also highlights the value of including multiple granularity levels in annotations, underscoring the need for such diversity to achieve a deeper understanding of video content.
3.    The authors perform a thorough analysis of the dataset structure, providing the research community with a clear picture of how this dataset challenges existing models. This comprehensive evaluation allows researchers to recognize the distinct aspects of the dataset that may stress-test model capabilities. By offering insights into the specific challenges posed by the dataset’s diverse annotations, the authors present a strong case for the benchmark’s relevance in advancing multi-modal video understanding, especially given its focus on varied text descriptions and granularities.
 4.    The study maximizes the potential of large language models (LLMs) and pre-trained models, creating robust annotations that are contextually rich and diverse. Leveraging these models has allowed the authors to generate a wide range of annotations that are both high-quality and informed by extensive world knowledge. This methodological approach is particularly commendable, as it goes beyond simple label generation and instead creates an annotation set that reflects nuanced information, adding substantial depth to the benchmark.

**Weaknesses:**

1.    While the dataset is a valuable addition to the field, offering some level of novelty, it falls short of significant innovation. The work’s main strength lies in diversifying existing benchmark annotations rather than introducing a groundbreaking methodology or framework. Although this level of novelty is acknowledged, the lack of an entirely new approach may limit the perceived impact of the benchmark within the research community. The work could benefit from a clearer distinction between its novel contributions and those of prior benchmarks to emphasize its unique value.
2.    The dense captioning provided by the dataset primarily centers on spatial relationships, which, while useful, presents a relatively narrow challenge to models. By focusing mainly on spatial relations, the annotations risk being overly uniform in terms of complexity, potentially limiting the depth of the benchmark’s impact on model evaluation. A more varied approach, incorporating a broader spectrum of semantic relationships beyond spatial ones, could introduce more diverse challenges for model training and testing, offering a more robust assessment of model capabilities.
3.    The paper does not introduce new baseline models, especially for dense caption-based tasks, to validate the proposed benchmark’s utility. The absence of baseline models that are specifically designed to adapt to the dense captioning structure leaves an open question about whether existing models are sufficient or if structural changes are required to adapt models without compromising previous task integrity. Designing or adapting baseline models could serve as a valuable proof of concept, demonstrating the benchmark’s effectiveness and providing the community with a clear starting point for future research on model adaptations for dense captioning.

**Questions:**

Given the paper’s use of LLM-based information, a question arises: why not expand the scope from single-object to multi-object tracking, a more semantically challenging task that could further elevate the benchmark’s utility? Multi-object tracking would offer a richer context for assessing model capabilities in complex, real-world scenarios and create additional opportunities to explore and refine model understanding in multi-modal video tracking.

---

### Official Review · Reviewer_J3Fx · 2024-11-02

**Soundness:** 3
**Presentation:** 3
**Contribution:** 3
**Rating:** 5
**Confidence:** 4

**Summary:**

The paper proposes a new visual language tracking benchmark featuring diverse texts, built upon five prominent VLT and SOT benchmarks. It is the first to leverage the extensive knowledge base of large language models (LLMs) to deliver multi-granularity and diverse semantic information. Experiments initially demonstrate the impact of diverse texts on tracking performance.

**Strengths:**

- The proposed DTVLT is the first comprehensive VLT benchmark using LLM to provide multi-granularity and diverse semantic information.

- The quantity of the dense texts in DTVLT is 45.9 times larger than the official annotations of previous datasets, which will highly support further research in VLT and video understanding.

- The paper is easy to follow and clearly states its contributions, providing a comprehensive presentation of the content, including bad cases.

**Weaknesses:**

- Tracking by natural language specification enhances human-machine interaction and has the potential to improve tracking outcomes. Although this paper introduces four levels of granularity in text annotations within the benchmark, the individual contributions of each text type to model performance remain ambiguous. As shown in Figure 5, models retrained on the DTVLT dataset exhibit improved performance on DTVLT; however, it is difficult to determine whether these models have merely memorized the patterns within DTVLT. Therefore, could you evaluate the models trained on DTVLT on an entirely different VLT dataset to assess model performance improvements? Additionally, could you train separate models on each of the four text granularity datasets and compare their performance to isolate the impact of each type?

- DTVLT provides significantly more diverse semantic information than previous datasets, with texts generated by large language models (LLMs), which differ from human language. Therefore, conducting real-world experiments to compare models trained with and without DTVLT is valuable for evaluating performance and robustness in practical applications. Have you considered conducting user studies or real-world deployment tests in specific domains like autonomous driving or surveillance to compare the performance of models trained with and without DTVLT?

- Considering the word distribution in DTVLT, as shown in Figure 4, terms like 'center', 'positioned', and 'black' constitute a large proportion, which may lead to bias in models trained on this dataset. It would be beneficial to analyze how this distribution impacts model performance. Could you conduct an ablation study to measure how removing or reducing the frequency of common terms like 'center', 'positioned', and 'black' affects model performance?

- In terms of bad cases, complex scenarios with multiple similar objects often pose difficulties. As shown in Figure A9, many objects match the detailed description, such as 'A zebra is standing in the middle of the field', which may make it harder for the model to accurately identify the target. Therefore, filtering out redundant information may enhance the quality of the dataset. Have you considered assessing the frequency of such ambiguous cases in the dataset and implementing a post-processing step to refine the generated descriptions?

**Questions:**

See the Weaknesses.

---

### Official Review · Reviewer_eR3b · 2024-11-02

**Soundness:** 3
**Presentation:** 3
**Contribution:** 3
**Rating:** 3
**Confidence:** 5

**Summary:**

The paper introduces a new benchmark, DTVLT, for Visual Language Tracking (VLT) that enhances traditional single-object tracking by integrating more diverse linguistic data through large language models (LLMs). Current VLT benchmarks rely on brief, standardized annotations, which limit the depth of video content understanding, leading algorithms to rely on memorization rather than genuine comprehension. By leveraging LLMs, DTVLT provides varied semantic annotations at multiple levels of detail for existing VLT and single-object tracking datasets, supporting tasks like short-term, long-term, and global instance tracking. This benchmark includes four text granularity levels, enabling richer semantic understanding by capturing broader context and subtle video content dynamics. Experimental analyses show that diverse text annotations improve tracking performance and reveal limitations in current algorithms, encouraging further VLT research.

**Strengths:**

1. The manuscript is well-structured, with clear, fluent writing that makes the content easy to understand.

2. The authors introduce a new benchmark, DTVLT, based on five prominent VLT and SOT benchmarks, addressing three tracking tasks: short-term tracking, long-term tracking, and global instance tracking. DTVLT provides four granularity combinations, capturing the extent and density of semantic information, and incorporates DTLLM-VLT, which uses LLM to generate diverse, high-quality language descriptions.

3. Comprehensive experimental analyses evaluate the effect of diverse textual descriptions on tracking performance, with insights into performance bottlenecks in existing algorithms that may aid future VLT research.

**Weaknesses:**

1. In Table 3, did the author use the original implementations and pretrained weights for these trackers (MMTrack, JointNLT, and UVLTrack)? If so, it would be helpful for the author to discuss any discrepancies between their results and those reported in the original papers, as the performance results for these trackers in Table 3 do not align with the performance reported in their respective publications.

2. As this paper proposes a benchmark, additional trackers should be tested to fully evaluate it (i.e., VLT ("Divert More Attention to Vision-Language Tracking")).

3. I am skeptical about the necessity of the "Initial Concise" annotation by the LLM. This annotation seems less effective than the official language, as the performance decreases when directly testing the trackers, as shown in Table 3. Additionally, Figure A7 indicates that even with retraining, performance still declines with the "Initial Concise" annotation. Could the author provide a more detailed analysis of why they included the "Initial Concise" annotation despite its apparent limitations, and what specific value it adds to the benchmark.

4. When directly testing tracking performance with "Dense Concise" or "Dense Detailed" annotations—such as for the N-th test frame in a video, which language annotation does the author use? How their choice of annotation for each frame might impact the tracking results, and whether they considered alternative approaches.

5. In the training process using "Dense Concise" or "Dense Detailed' annotation, the model needs sample the N-th training frame from the given video. Which language annotation does the author use in this case. how the choice of annotation during training might affect the model's ability to generalize to different text granularities during testing.

6. In Figure A7, the performance with "Dense Detailed" annotation appears worse for the AUC metric. Could the author provide a hypothesis for why the "Dense Detailed" annotation leads to worse AUC performance, and to discuss the implications of this finding for the design of VLT systems.

**Questions:**

Please refer to the concerns and issues raised in the "Weaknesses".

**Details Of Ethics Concerns:**

This manuscript may violate the anonymity policy, as the authors mention in lines 242-243: 'To overcome these challenges, we have developed DTLLM-VLT (Li et al. (2024a)),.

---

### Official Review · Reviewer_Xyaf · 2024-11-03

**Soundness:** 2
**Presentation:** 2
**Contribution:** 2
**Rating:** 5
**Confidence:** 3

**Summary:**

This paper presents DTVLT, a new multi-modal benchmark for Visual Language Tracking (VLT), aiming to enrich existing VLT datasets by introducing diverse textual annotations generated by large language models (LLMs). DTVLT is built upon five existing VLT and SOT benchmarks, and it employs a multi-granularity text generation strategy to provide descriptions of varying lengths and detail levels. The benchmark is designed to support three main tasks—short-term tracking, long-term tracking, and global instance tracking—providing a more comprehensive environment for evaluating VLT algorithms.

**Strengths:**

(1) Enhanced Text Diversity: The benchmark introduces a multi-granularity approach to text descriptions, which offers different semantic densities. This allows for a finer evaluation of VLT algorithms' ability to handle both concise and detailed language input.
(2) Integration with LLMs: By leveraging LLMs, the paper automates text generation, potentially reducing the need for costly manual annotations and allowing for large-scale text generation across various scenarios.
(3) Comprehensive Experimental Setup: The paper evaluates multiple VLT models on DTVLT, providing a detailed analysis of how different text granularities impact tracking performance. This analysis can offer insights into the limitations of current VLT models.

**Weaknesses:**

(1) This paper appears more like a pipeline proposal, merely leveraging existing models to enhance an existing dataset, without focusing on the scientific questions that should be addressed when establishing this dataset.
(2) The motivation is unclear, as Dtllm-vlt[1] seems to have already addressed the short-term tracking, long-term tracking, and global instance tracking challenges mentioned in this paper.
(3) DTVLT heavily relies on LLM-generated text descriptions but lacks a quality control method for these generated texts. The automatically generated text may contain semantic biases, inconsistencies, overly long or redundant information, and may even be disconnected from the visual content, particularly in multi-modal tasks. Analyzing the quality of generated texts and investigating how to produce better descriptions would be more worthwhile.

[1] DTLLM-VLT: Diverse Text Generation for Visual Language Tracking Based on LLM.

**Questions:**

(1) What are the specific scientific challenges this dataset aims to address, beyond what existing benchmarks like Dtllm-vlt have already solved?
(2) What quality control measures were implemented to ensure that LLM-generated texts accurately describe the visual content, and how were issues like semantic bias or redundancy managed?
(3) For additional questions, please refer to the Weaknesses section.

---

### Note · Authors · 2024-11-12

I have read and agree with the venue's withdrawal policy on behalf of myself and my co-authors.